# Solid Lipid Nanoparticles Enhancing the Leishmanicidal Activity of Delamanid

**DOI:** 10.3390/pharmaceutics16010041

**Published:** 2023-12-27

**Authors:** Javier Santamaría-Aguirre, Daniela Jacho, Miguel A. Méndez, Ana Poveda, Javier Carrión, Mónica L. Fanarraga

**Affiliations:** 1Departamento de Biología Molecular, Universidad de Cantabria, 39011 Santander, Spain; 2Grupo de Nanomedicina, Instituto Valdecilla—IDIVAL, 39011 Santander, Spain; 3Grupo de Investigación en Biodiversidad, Zoonosis y Salud Pública (GIBCIZ), Instituto de Investigación en Zoonosis (CIZ), Facultad de Ciencias Químicas (FCQ), Universidad Central del Ecuador, Quito 170521, Ecuador; 4Facultad de Ciencias Químicas (FCQ), Universidad Central del Ecuador, Quito 170521, Ecuador; 5Departamento de Ingeniería Química, Instituto de Simulación Computacional (ISC-USFQ), Universidad San Francisco de Quito USFQ, Quito 170157, Ecuador; 6Grupo ICPVet, Departamento Sanidad Animal, Universidad Complutense de Madrid, 28040 Madrid, Spain; 7Instituto de Investigación Hospital 12 de Octubre, 28041 Madrid, Spain

**Keywords:** zoonosis, trypanosomatid, nanomedicine, neglected, fluoroquinolone, benzimidazoles

## Abstract

Leishmaniasis, a zoonotic parasitic disease transmitted by infected sandflies, impacts nearly 1 million people yearly and is endemic in many countries across Asia, Africa, the Americas, and the Mediterranean; despite this, it remains a neglected disease with limited effective treatments, particularly in impoverished communities with limited access to healthcare. This study aims to repurpose approved drugs for an affordable leishmaniasis treatment. After the screening of potential drug candidates by reviewing databases and utilizing molecular docking analysis, delamanid was chosen to be incorporated into solid lipid nanoparticles (SLNPs). Both in cellulo and in vivo tests confirmed the successful payload release within macrophages and through the epidermis following topical application on murine skin. The evaluation of macrophages infected with *L. infantum* amastigotes showed that the encapsulated delamanid exhibited greater leishmanicidal activity compared with the free drug. The process of encapsulating delamanid in SLNPs, as demonstrated in this study, places a strong emphasis on employing minimal technology, ensuring energy efficiency, cost-effectiveness, and reproducibility. It enables consistent, low-cost production of nanomedicines, even on a small scale, offering a promising step toward more accessible and effective leishmaniasis treatments.

## 1. Introduction

Leishmaniasis is a neglected tropical parasitic disease that predominantly affects impoverished communities in tropical areas, placing a disproportionate burden on women and children. Factors such as poverty, malnutrition, limited healthcare access, and population displacement exacerbate the situation [1,2,3,4]. The disease is caused by various *Leishmania* protozoa species, spreads through the bites of infected female sandflies [5], and can manifest in three forms, ranging from skin lesions to a nearly always fatal visceral type. Leishmaniasis is endemic in over 90 countries across tropical and subtropical regions of Asia, Africa, the Americas, and the Mediterranean. However, changes in temperature and rainfall patterns due to climate change are influencing the size and distribution of sandfly populations [6,7].

Every year, an estimated 700,000 to 1 million new cases of leishmaniasis emerge and impose devastating health, social, and economic consequences on over one billion people according to the World Health Organization [8]. These challenges are further intensified by shifts in demographics, leading to increased vulnerability to various hardships, including high healthcare costs, productivity loss [9], disability, stigma [10,11], and discrimination, primarily due to the nature of the lesions. Unfortunately, despite these alarming statistics, there has not been adequate public and private investment in prevention and control measures [6].

Treatment of leishmaniasis is challenging in poor communities with limited access to regular health care. The choice of treatment for leishmaniasis depends on the specific clinical presentation, which is further influenced by the parasite species and geographical distribution. The efficacy of treatment is primarily influenced by the level of toxicity and associated side effects, which can affect the willingness of patients to undergo treatment. In addition, the drugs used can be both expensive and in short supply [12]. The Pan American Health Organization recommends the use of pentavalent antimonials, such as meglumine antimoniate, administered once daily for up to 30 days. Regrettably, the associated adverse effects, including myalgia and arthralgia, often lead to non-compliance and contribute to the development of resistant strains [13,14].

The first stage of this study is dedicated to examining existing approved drugs for their potential applicability in treating leishmaniasis through repositioning [15,16,17]. This approach provides notable benefits in terms of both time and cost efficiency compared with the development of entirely new compounds [18,19]. Certain approved and previously discarded molecules may suffer from issues like inadequate solubility, restricted bioavailability, undesirable side effects, and dosing complexities, all of which can potentially be mitigated through the process of nanoencapsulation [20,21]. 

Among the various available nanoencapsulation systems, solid lipid nanoparticles (SLNPs) present a promising platform for drug delivery [22,23,24,25]. They enhance the solubility and bioavailability of hydrophobic drugs, provide sustained release for prolonged therapeutic effects, and improve chemical stability [26,27]. Additionally, SLNPs boast benefits like better biocompatibility, targeted delivery, reduced toxicity, and protection from enzymatic degradation, making them versatile for pharmaceutical applications. They also hold potential for efficient production processes and minimization of variability [10].

In the context of leishmaniasis, as well as other neglected tropical diseases, the successful use of SLNPs depends on the streamlined formulation and production of methods that are cost-effective, time-efficient, and environmentally responsible [28,29,30,31]. This approach should address supply concerns: the industrial-scale production of nanoparticles implies high technology costs for manufacturing, storage, and distribution; the small-scale, on-demand, and personalized production of nanomedicines can improve their accessibility, especially for impoverished populations.

To address these needs, our research begins by identifying a leishmanicidal compound from a pool of drugs through database analysis and molecular docking. After confirming the compound’s effectiveness against the *Leismania* promastigote forms, we proceed to create a point-of-care-compatible SLNP formulation using the solvent–antisolvent method to encapsulate the drug. Subsequently, we evaluate the formulation in vitro and in cellulo, ultimately demonstrating its efficacy against the amastigote stage.

## 2. Materials and Methods

### 2.1. Identification of Compounds Exhibiting High Activity against Trypanosomatids and Minimal Cytotoxicity

#### 2.1.1. Database Processing

The information was obtained from the databases of “Pathogen box” (https://www.mmv.org/mmv-open/pathogen-box/about-pathogen-box (accessed on 1 January 2019) from the initiative Medicines for Malaria Venture (MMV: https://www.mmv.org/about-us/what-we-do/overview-our-work (accessed on 1 January 2019)). The data were complemented with scientific articles and web pages, which allowed the preparation of a table showing activity against *T. cruzi*, *T. brucei*, *L. infantum*, and *L. donovani*, cytotoxicity in macrophages and hepatocytes, as well as physicochemical parameters. We classified the data into three levels, assigning each substance a score that served to prioritize the substances with the greatest possibility of presenting leishmanicidal activity.

#### 2.1.2. Molecular Docking

Molecular docking was performed using Auto Dock 4.0 Vina software to analyze the proteins DHFR-TS and topoisomerase II (TOP II) from Leishmania. The protein structures were obtained from PubChem, and their pKa and pH-dependent distribution curves were calculated using ChemAxon’s Chemicalize tool http://www.chemaxon.com (accessed on 15 November 2023), Budapest—Hungary. The charges of the molecules were adjusted for pH 7 using AC/DC 12.0 and MarvinSketch software 5.0. The modified structures were debugged, optimized, and corrected using the Check Structure tool. Human alpha and beta TOP II proteins and TOP II from *L. amazonensis* were selected based on previous research by Uzcanga et al. (2017) [32] and Espíndola (2019) [33], respectively. The Docking Server was utilized to test the best binding sites identified previously. Forty-two fluoroquinolones and meglumine antimoniate were used as ligands. AutoDock 4.0 tools were also employed. A more detailed procedure is described in the Appendix A [33,34].

### 2.2. Determination of the Leishmanicidal Activity of the Selected Drugs

#### 2.2.1. Leishmanicidal Activity in Promastigotes

*L. infantum* (M/CAN/ES/96/BCN150 zymodeme MON-1) promastigotes were maintained for 5–7 days in Schneider medium supplemented with 10% FBS; then, 1 × 10^5^ promastigotes were seeded in microcentrifuge tubes with 1 mL of medium. The effect of the drug or nanoparticles was compared with the controls after 5 days of incubation at 25–27 °C. Parasite viability was performed in a hemocytometer. Assays were performed in triplicate.

A fluorescence method described previously [35] was applied to drug substances, (Appendix A). Briefly, 10^4^ promastigotes were cultured for 72 h at 25 °C in the presence/absence of the drug dissolved in the medium at the indicated dilutions. Hypotonic lysis buffer (20 mM Tris pH: 7.5, 5 mM EDTA, 0.008% Triton, 0.008% Sarcosine) with a fluorophore (Sybrgold 0.1×) was added. Fluorescence was measured with excitation at 495 nm and emission at 539 nm. To determine cytotoxicity, the same procedure was followed using J774 macrophages in DMEM supplemented with 10% FBS, which was incubated at 37 °C with 5% CO_2_. 

The leishmanicidal activity of drugs and nanoparticles was also measured using flow cytometry. Promastigote cultures of *L. infantum*, *L. major* (clone V1:MHOM/IL/80/Friedlin), and *L. mexicana* cultivated for 5–7 days were fixed using 4% paraformaldehyde, followed by thorough washing. Subsequently, they were stained overnight with a 1:500 dilution of propidium iodine. A total of 10,000 events minimum were processed in each flow cytometry experiment. Assays were performed in triplicate.

#### 2.2.2. Phase-Contrast Microscopy Imaging

Live promastigotes were photographed with a Nikon Eclipse TS 100 phase contrast microscope using the 10× and 20× lenses.

### 2.3. Delamanid Encapsulation in SLNPs

#### 2.3.1. Design and Synthesis of the SLNPs

Delamanid (CAS 681492-22-8, TargetMol), stearic acid (CAS 57-11-4, Thermo Scientific, Waltham, MA USA), and the ammonium salt of N-palmitoyl homocysteine, and PHC (CAS 474942-73-9, Avanti Polar lipids, Drive Alabaster, AL, USA) were solubilized in ethanol (Code 141086, PanReac AppliChem, Chicago, IL, USA). Cetyltrimethylammonium bromide, CTAB (CAS 57-09-0, Thermo Scientific), Tween 80 (CAS 9005-65-6, Sigma Aldrich, St. Louis, MO, USA), and phosphate buffer saline (PBS) (Ref. 70011-036, Gibco, Waltham, MA, USA) were dispersed in a microcentrifuge tube. The organic phase was injected into the aqueous phase, sonicated (5 cycles 30 s each) at 130 W/20 KHz, filtered through a membrane, and stored at 4 °C until use.

For analytical purposes, commercially available magnetic nanoparticles (BNF Dextran, Product code 84-00-102, Micromod, Wallington, CT, USA) were introduced into the aqueous phase of the SLNP formulation; its original size, 100 nm, is fragmented during sonication (see below). This inclusion aided in the separation of SLNPs through centrifugation. The incorporation of magnetic nanoparticles did not show any noticeable effect on the leishmanicidal activity as demonstrated in Appendix A.

#### 2.3.2. Characterization of SLNPs

The SLNP dispersion was diluted in PBS and measured via Dynamic Light Scattering (DLS) in a Malvern Zetasizer Ultra. Transmission Electron Microscopy (TEM) imaging was performed in a Jeol JEM 1011 equipped with a Gatan Orius Sc 1000 CCD, using SLNP placed and dried on a grid; no contrast was necessary since fragments of magnetic nanoparticles were included in the formulation for this purpose. Drug encapsulation efficiency was determined indirectly by quantifying the non-encapsulated drug in the supernatant, measuring its absorbance at 340 nm, and using a calibration curve.

#### 2.3.3. In Vitro Drug Release

SLNP dispersions in PBS were subjected to agitation, 200 rpm at 37.5 °C for 8 days, to investigate in vitro drug release. The absorbance of 500 µL samples from the supernatant was determined at 340 nm, and the quantification of the drug was performed employing a calibration curve.

### 2.4. In Cellulo Activity of SLNPs

#### 2.4.1. Cellular Proof-of-Concept Drug Release

SLNPs carrying both DiI (41085-99-8, Invitrogen, Carlsbad, CA, USA) were prepared as a proof-of-concept (PoC) to investigate drug release in cellulo. After 24 h of exposure to these nanoparticles, J774 macrophages were fixed using 4% paraformaldehyde. They were then stained with DAPI (CAS 28718-90-3, Sigma Aldrich) and mounted on slides for imaging using a Nikon A1R confocal microscope.

#### 2.4.2. Transdermal PoC Drug Release

Eight-week-old female and male BALB/c mice (Janvier-Labs, Laval, France) were maintained in the Animal Facility of the Complutense University of Madrid under specific pathogen-free conditions without food or water restriction. The animal research described in this manuscript complied with Spanish (Ley 6/2013) and European Union legislation (2010/63/UE). The protocols used were approved by the Animal Care Committee of the Complutense University of Madrid. The animals were carefully shaved to facilitate the application of 20 μL of the dispersion containing SLNPs loaded with both the drug and DiR (CAS 100068-60-8, Invitrogen). The treatment was left in place as indicated in the text. After humane sacrifice, the skin was dissected, fixed in 4% formalin, and embedded in paraffin. The samples were then sectioned, stained, and prepared for confocal microscopy imaging.

#### 2.4.3. Electron Microscopy Imaging of Infected Macrophages

Canine macrophage DH82, cultured in RPMI (RPMI Medium 1640 (1X) + Glutamax-l. Gibco. Ref. 61870-010) supplemented with 10% FBS and 0.05% Gentamicin, and incubated at 37 °C/5% CO_2_ were infected with *L. infantum* promastigotes for 4 h. Then, non-internalized parasites were removed and cells were treated. After 12 h, treatments (drug or SLNPs) at a concentration corresponding to the IC50 were added and incubated for 48 h. After treatment, the cells were fixed with glutaraldehyde 3% for inclusion and post-fixed in 2% buffered osmium tetroxide, dehydrated in a graded acetone series, and embedded in Araldite resin (Durcupan ACM, Sigma-Aldrich, St. Louis, MO, USA). Ultrathin sections of ca. 70 nm thick were obtained on an LKB ultramicrotome. The sections were stained with lead citrate and uranyl acetate. Imaging was performed using a JEOL JEM 1011 operated at 100 kV.

#### 2.4.4. Leishmanicidal Activity in Amastigotes

DH82 cells were infected with *L. infantum* live parasites on cell culture slide chambers for 4 h. The cells were treated with the drug, SLNPs, or control media for 48 h. The number of infected macrophages and amastigotes by cell was determined in 400 cells under optic microscopy using an Olympus BX41 on Giemsa-stained slides. The infection index was obtained by multiplying the number of infected cells by the number of amastigotes per cell. A one-way ANOVA statistical test, followed by post hoc comparisons using the Tukey HSD test, was used to evaluate differences.

The in vitro experiments were carried out in a laboratory equipped with laminar flow cabinets adapted for biosafety level 2 microorganisms following the procedures of Biosecurity Standard Operating Procedures (SOPs) applied to the Faculty of Veterinary Medicine.

## 3. Results

### 3.1. Identification of Compounds Exhibiting High Activity against Trypanosomatids and Minimal Cytotoxicity

#### 3.1.1. Database Processing

We assessed the activity of compounds against trypanosomatids and their cytotoxicity against macrophages using the “Medicines for Malaria Venture (MMV) project’s Pathogen box” database. The results were summarized in comparative tables for evaluation; briefly, the leishmanicidal activity values were stratified into tertiles and color-coded as green, orange, or red, corresponding to scores of 3, 2, and 1, respectively. The highest score was assigned to the lowest IC50 values, indicating superior leishmanicidal activity. Conversely, for cytotoxicity assessment, higher IC50 values were associated with elevated scores. To establish prioritization, these scores were multiplied to generate a priority number. Substances falling within the first tertile, marked by the highest combined scores, were considered to exhibit greater potential. In cases in which data were unavailable for promising molecules, we supplemented it with information from scientific articles. To refine the selection process, we took into account physicochemical parameters that are indicative of biological activity (drugability). These data were then cross-referenced with scientific literature, which offered valuable insights into potential targets and mechanisms of action. Finally, six different molecules were selected: MMV688262 (delamanid), MMV688372, MMV689061, MMV689437, MMV690102, and MMV690103 (Appendix A).

#### 3.1.2. Molecular Docking

Using Autodock Vina (https://vina.scripps.edu/ (accessed on 15 November 2023) (which is part of suite Auto dock 4.0), we conducted bioinformatic studies examining 42 fluoroquinolones representing all generations. Among them, compounds with high affinity for topoisomerase II (TOP II) of *Leishmania* and Salmonella Gyrase, coupled with low affinity for human TOP II alpha and TOP II beta, were identified. In the process of data analysis, affinity values were categorized into tertiles and color-coded as green, orange, or red, with corresponding scores of 3, 2, and 1, respectively. The highest score was attributed to the compounds displaying the greatest affinity for parasite enzymes. In contrast, for human enzymes, higher scores were linked to lower affinity values. To facilitate prioritization, these scores were multiplied to generate a comprehensive priority number. Substances situated within the first tertile, characterized by the most favorable combined scores, were deemed to possess superior potential. These molecules include trovafloxacin, tosufloxacin, and sitafloxacin. Enrofloxacin was selected for inclusion based on its previous use in in vitro research, along with its high bioavailability and low toxicity [36]. Furthermore, prulifloxacin (Appendix A) was included in the study due to its structural feature of a four-membered ring containing a sulfur atom, which has been reported to enhance its activity [37,38].

### 3.2. Assessing Leishmanicidal Activity of Selected Drugs against Promastigotes

#### 3.2.1. Leishmanicidal Activity in Promastigotes

Leishmanicidal activity assays were conducted in multiple phases, systematically comparing the efficacy of the experimental compound with that of ciprofloxacin. The effectiveness of the latter had been previously confirmed using fluorescence-based methodologies described previously [35]. Compounds exhibiting significant activity were subjected to dilution, and the test was reiterated until the concentration causing a fifty-percent reduction in the population (IC50) could be approximated. Delamanid displayed notable activity against *L. mexicana* promastigotes and prulifloxacin was the best among the fluoroquinolones (Appendix A).

To gain deeper insights into the prior findings, we applied the fluorescence method outlined in Section 2.2.1 to ascertain the IC50 values in promastigotes. The results indicated an IC50 value of 0.026 μM for delamanid, 180.2 μM for prulifloxacin, and 451.3 μM for enrofloxacin. Enrofloxacin was selected as a reference due to its previous usage on promastigotes, both in solution [39] and encapsulated in nanosystems [40,41].

Additionally, we employed flow cytometry to determine the IC50 (Table 1) for different *Leishmania* species and macrophage cell lines, DH82 and J774, and both were used in the in cellulo assays.

The IC50 of delamanid against the three *Leishmania* species is within the same order of magnitude, while the cytotoxicity in the assayed eukaryotic cells is at least 10,000-fold higher. These results agree with previous investigations that revealed an IC50 of 18.5 nM in *L. donovani* promastigotes derived from clinically isolated strains [42]. Given the significant difference in leishmanicidal activity observed between delamanid and prulifloxacin, coupled with its safety margin in eukaryotic cells, the study aimed toward delamanid (Figure 1a).

#### 3.2.2. Phase-Contrast Microscopy Evaluation of the Effect of Delamanid on Promastigotes

We utilized phase-contrast microscopy to assess the impact of delamanid on the parasite, focusing on potential morphological alterations and signs of cell death. This approach enabled us to analyze how delamanid influenced the growth and structure of the parasite (Figure 1b). Alongside a notable decrease in the population of *L. infantum* promastigotes, we noted morphological abnormalities such as an increase in their size, a more rounded shape, and noticeable cytokinetic abnormalities, all evident six days after a single administration of delamanid at its IC50 concentration.

### 3.3. Encapsulation of Delamanid in SLNPs

#### 3.3.1. Design and Synthesis of the SLNPs

The SLNP formulation comprises seven components; the exact quantities employed for the synthesis are shown in Table 2. The ethanol content is maintained well below the maximum allowable limit, the combined total of surfactants is kept below 0.12%, and the lipids are limited to 0.018%. The lipids complement each other due to their charges, and stearic acid is a widely recognized ingredient in the pharmaceutical and cosmetic industries [43]. The added delamanid corresponds to 27.7 μM in the SLNP dispersion.

The manufacturing process, as illustrated in Figure 2a, was intentionally designed to involve a minimum number of steps and incur low energy costs. One notable feature of this method is the remarkable consistency in the batches of nanoparticles produced, as demonstrated in Appendix A.

#### 3.3.2. Characterization of SLNPs

As shown in Figure 2b,d, the particle size does not exceed 200 nm with a highly reproducible polydispersity index. Four consecutive batches, prepared independently, showed minimal variation in particle size between 140–180 nm diameter and a polydispersity index of less than 0.25. The zeta potential measured approximately +10 mV, indicating a slight positive charge (refer to Appendix A). Importantly, the encapsulation of delamanid in the SLNPs did not alter their sizes or zeta potential. The stability of the physical characteristics of the SLNPs was assessed for 7 days at 4 °C (Appendix A) and no discernible changes in properties were observed.

Regarding drug encapsulation, the produced SLNPs demonstrated an average encapsulation efficiency of 84.5% (Appendix A). When it comes to drug release from SLNPs under physiological conditions (pH 7.4, 37 °C), these nanosystems exhibited a sustained release pattern for delamanid. Specifically, over half of the content was released during the first four days. The 80% threshold was surpassed by the sixth day (Figure 2c).

### 3.4. SLNP Payload Release: Proof-of-Concept

#### 3.4.1. In Cellulo and In Vivo Payload Release

Next, we exposed macrophage cells to the SLNPs in the culture media to examine the uptake, the pathways for intracellular nanoparticle transport, and the payload release. To assess the effectiveness of intracellular drug release within macrophages following SLNP internalization, we conducted a proof-of-concept experiment by incorporating a fluorescent pigment into the delamanid-loaded SLNPs (Appendix A). After 24 h of exposure to the nanoparticles, macrophages were imaged using confocal microscopy. The study revealed a prominent fluorescence in their cytoplasms, indicative of the successful release of the nanoparticle payload intracellularly (Figure 3a, Inset #1). This confirms both the capture of the SLNPs and the release of the cargo.

Finally, to evaluate the in vivo payload release, we conducted a proof-of-concept experiment to evaluate the release of the compounds. In this experiment, SLNPs loaded with both the fluorescent dye and delamanid were applied topically on normal murine skin. Skin from treated mice was collected at various times after topical application and histologically processed for confocal microscopy imaging to assess compound release and diffusion through different skin layers. Figure 3b–d depicts different time points at which the skin was observed to assess the penetration of the dye (indicated by arrows). Immediately after the initial application, the fluorescent dye was primarily observed within the corneal stratum (Figure 3b). Two hours later, the dye had successfully penetrated down to the basal cell layer of the dermis (Figure 3c). After 24 h, the dye was mainly concentrated in the hair follicles (Figure 3d).

These findings illustrate that the nanoparticles demonstrate a high level of efficiency in releasing their cargo at the intracellular level. Additionally, they highlight the potential of the nanoencapsulation system for transdermal delivery of lipophilic molecules like delamanid. Moreover, because of its fully biocompatible formulation, there is optimism regarding the efficient delivery of the treatment through alternative routes of administration.

#### 3.4.2. Subcellular Distribution of SLNPs in Infected Macrophages

To study the entry of SLNPs into macrophages, the nanoparticle subcellular pathway, and the effect of amastigotes, we infected macrophages with *L. infantum* for 48 h. Following the incubation period, the parasites were removed by washing, and the macrophages were then treated with SLNPs. For imaging purposes, magnetic nanoparticles were incorporated into these nanoparticle batches. Following an additional 48 h exposure to the SLNPs, the cells were fixed, dehydrated, and embedded in resin for ultramicrotomy and TEM imaging.

Figure 4 shows a section of two representative infected macrophages. Some intracellular nanoparticles were observed surrounded by membranes, probably phagosomes (Figure 4, inset # 5a), suggesting uptake by phagocytosis, while others were also seen free in the cytoplasm of cells, lacking surrounding membranes (Figure 4, insets #5b, #6). The distinct recognition of SLNPs is facilitated by the inclusion of iron nanoparticles in the formulation, making their structure visible and clearly distinguishable from other cellular structures observed in these sections.

These findings are consistent with confocal microscopy observations and suggest successful uptake of SLNPs by macrophages, the cells targeted for treatment. The results suggest that this novel formulation shows promise for the treatment of leishmaniasis.

#### 3.4.3. Comparative Leishmanicidal Efficacy of SLNPs against Intracellular Amastigotes

To assess the effectiveness of SLNP treatment compared with the controls, canine macrophages infected with *L. infantum* were subjected to three different treatments: (i) unloaded nanoparticles as a control (SLNP), (ii) nanoparticles encapsulating delamanid (SLNP Dm), and (iii) the free drug (Dm). Following treatment, the number of amastigotes per infected cell was quantified on Giemsa-stained cells. The resulting values were used to calculate the infection index, providing an overall measure of parasite burden (Figure 4b).

A one-way ANOVA statistical test was performed to assess the impact of treatments on the Infection Index of DH82 macrophages. The analysis revealed a statistically significant effect of the treatment on the Infection Index at the *p* < 0.05 level, compared with the control conditions [F(2, 15) = 22.13, *p* < 0.0001]. Post hoc comparisons using the Tukey HSD test indicated that the mean score for SLNP Dm (M = 30.95, SD = 25.36) was significantly different from SLNP (M = 147.12, SD = 30.34) and Dm (M = 77.20, SD = 34.92). 

These results demonstrate that the delamanid encapsulated in SLNPs displays a markedly enhanced leishmanicidal effect when compared with delamanid in its unencapsulated form, particularly in its potency against intracellular amastigotes of *L. infantum*.

## 4. Discussion

Delamanid, an approved oral treatment for multidrug-resistant *Mycobacterium tuberculosis*, possesses high lipophilicity, making it compatible with the lipid-rich matrix of SLNPs. These nanoplatforms offer a range of benefits, particularly in overcoming biological barriers and providing sustained drug release. Notably, the manufacturing process is highly versatile, allowing for various administration routes, including topical, parenteral, inhalation, and even routes like ocular, oral, or even intra-cerebral administration [44].

In comparison with other nanoencapsulation methods, which often necessitate the application of thermal or mechanical energy (such as heating, shearing, ultrasound, etc.) or induce nanoparticle formation through alterations in solubility and surface tension [44,45,46], the approach employed in this study is notably streamlined and straightforward. It combines principles like solvent–antisolvent insolubilization, ultrasound-mediated top-down processes, and reduction in interfacial tension. Importantly, this method can be carried out without the need for high-tech equipment. As a result, it yields SLNPs with highly consistent and reproducible characteristics across different synthesis batches.

The mechanism of action of delamanid on *Mycobacterium* may entail disrupting the biosynthesis of crucial components of the cell wall, namely methoxy and keto mycolic acids. Additionally, it may interfere with respiratory activity in the bacteria [47]. However, it is important to note that mycolic acids, which are commonly encountered in the cell envelopes of various microorganisms, are absent in *Leishmania* [48]. However, this parasite does contain cis-9,10-methylene octadecanoic acid, a fatty acid structurally akin to mycolic acids. This similarity suggests that it could potentially be the target of delamanid’s leishmanicidal activity [49]. It is worth noting that a higher content of C19∆ has been associated with resistance to antileishmania drugs like amphotericin B and miltefosine [50]. Additionally, cyclopropane fatty acid synthase null mutants of *L. mexicana* display altered cell shape, increased sensitivity to acidic pH, and modified growth patterns in serum-free media [51].

It is also important to consider the potential involvement of Aurora kinases as targets for delamanid in *Leishmania*. The significance of hesperidin, an inhibitor, in preserving cytoskeletal integrity and facilitating cytokinesis, as indicated by previous research [52], could potentially offer insight into the observed experimental outcomes. Similar results were replicated with inhibitors such as Barasertib and GSK-1070916 [53].

## 5. Conclusions

This study primarily focused on formulating SLNPs for the delivery of delamanid in the treatment of leishmaniasis. The process encompassed the selection of delamanid among different drug candidates, its incorporation into SLNPs, and subsequent evaluation of its effectiveness against *Leishmania* parasites. Encapsulation efficiency was notably high, and the released drug exhibited prolonged release characteristics. The results showcased successful payload release within macrophages, as well as efficient skin penetration following topical application. Moreover, delamanid demonstrated heightened activity against intracellular amastigotes when incorporated into the SLNPs formulated in this study. The manufacturing process proved to be not only straightforward and energy-efficient but also cost-effective and highly reproducible. In summary, this research suggests that delamanid-loaded SLNPs hold substantial promise as an effective treatment for leishmaniasis, representing a significant advancement in the quest for improved therapies against this parasitic disease.

## Figures and Tables

**Figure 1 pharmaceutics-16-00041-f001:**
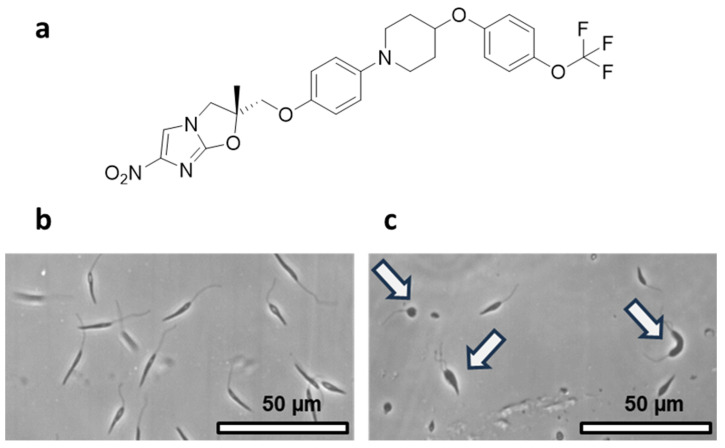
Molecular structure of delamanid and impact on promastigotes. (**a**) Chemical structure of delamanid, a bicyclic nitroimidazole compound approved for the treatment of multi-drug resistant tuberculosis, exhibiting high activity against *Leishmania* parasites. (**b**) Untreated *L. infantum* promastigotes after 7-day culture. (**c**) Morphological alterations induced by delamanid in *L. infantum* promastigotes after 2 days of treatment.

**Figure 2 pharmaceutics-16-00041-f002:**
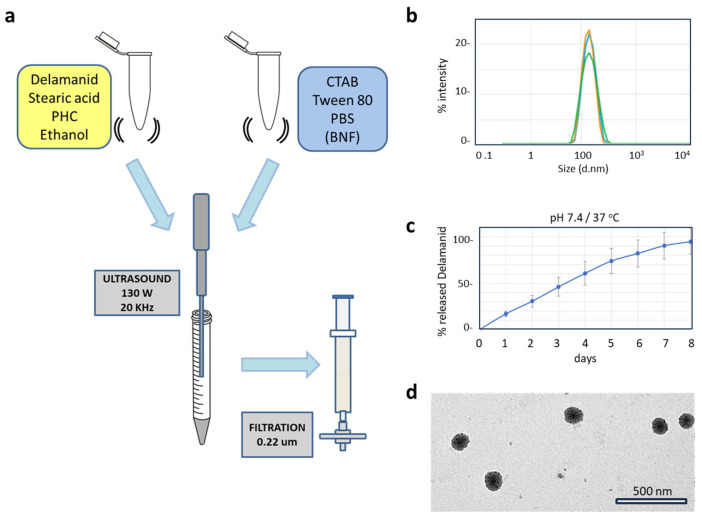
SLNP synthesis and characterization. (**a**) Schematic representation of SLNP synthesis using the solvent–antisolvent method. (**b**) Characterization of hydrodynamic diameter. Data from three distinct batches (in blue, green and orange colors) exhibit high reproducibility in particle size, as determined using Dynamic Light Scattering (DLS), with a polydispersion index below 0.25. (**c**) In vitro drug release profiles indicate sustained release over one week. (**d**) The SLNP dispersion was diluted at a ratio of 1 to 10 in PBS, applied to a grid, and left to air-dry at room temperature overnight before observation under TEM.

**Figure 3 pharmaceutics-16-00041-f003:**
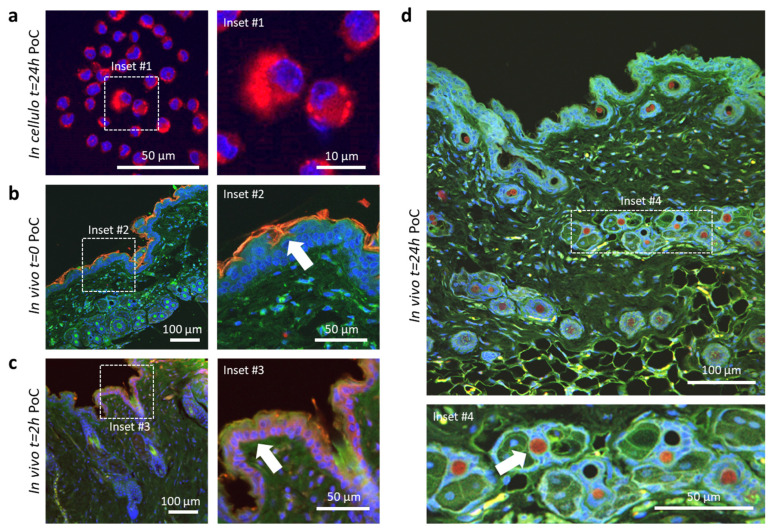
Liberation of payload from SLNPs containing dye and delamanid. (**a**) After 24 h of exposure to SLNPs, the macrophages display robust cytoplasmic fluorescence, indicating the successful release of the encapsulated dye intracellularly (Inset #1). (**b**–**d**) Confocal microscopy images of skin sections taken at sequential time points after the topical administration of the SLNPs show distinct patterns. (**b**) Immediately after application, the released compound is visible in the corneal stratum (Inset #2, arrow). (**c**) In 2 h, the dye penetrated the basal layer of the dermis, which was visible intracellularly (Inset #3, arrow). (**d**) A notable concentration of the dye around the hair follicles is detected at 24 h. The red channel shows the fluorescence of the dye (Inset #4, arrows). Nuclei were stained with Dapi (blue channel). The green channel represents tissue autofluorescence.

**Figure 4 pharmaceutics-16-00041-f004:**
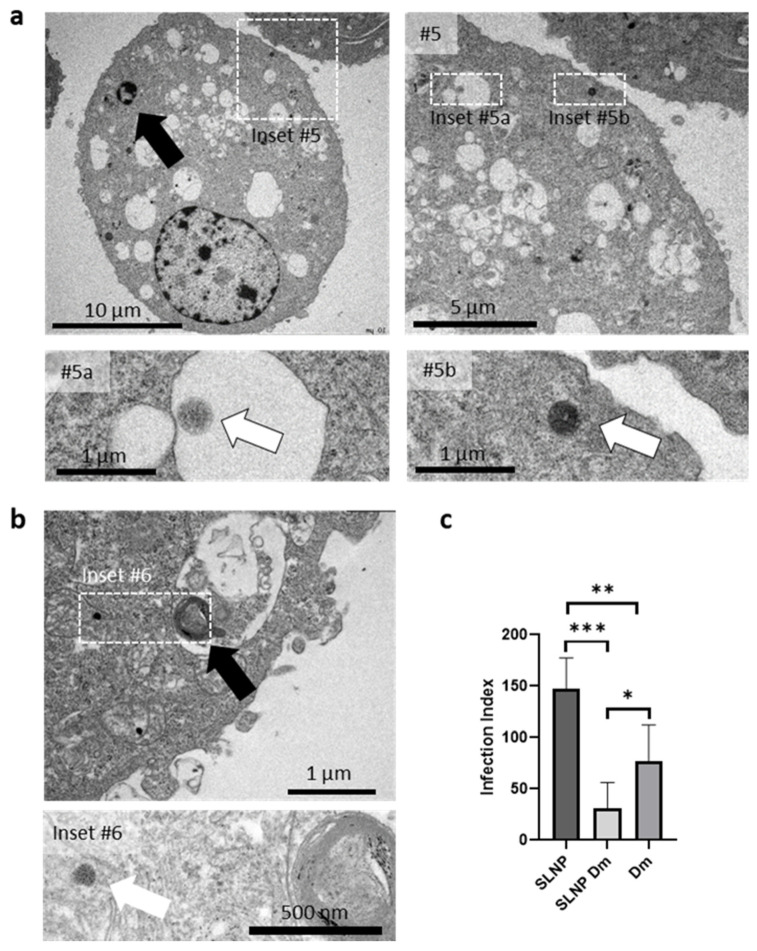
Effect of SLNP Treatment on Amastigotes. (**a**,**b**) Subcellular localization of SLNPs in infected macrophages. TEM images of an ultrathin section of an infected macrophage treated with SLNPs. Black arrows delineate the localization of intracellular dead parasites. Insets #5a and #5b offer magnified views of the areas enclosed in Inset #5. Inset #6 highlights the specified region in (**b**). White arrows indicate intracellular SLNPs, some observed within a putative phagosome (Inset #5a), while others are freely present in the cytoplasm (Insets #5b, #6). The presence of magnetic nanoparticles inside serves as the identifier for the SLNPs. (**c**) The use of SLNPs containing delamanid (SLNP Dm) significantly reduces the Infection Index of macrophages infected with *L. infantum* in comparison with the control groups, which include unloaded SLNPs and the plain drug (Dm). * *p* < 0.05, ** *p* < 0.01, *** *p* < 0.001.

**Table 1 pharmaceutics-16-00041-t001:** Determination of delamanid IC50 against leishmania species and macrophages using flow cytometry (n = 3, media ± SD).

Species/Cell Line	IC 50 (nM)
*L. infantum*	17.2 ± 0.9
*L. major*	21.9 ± 2.8
*L. Mexicana*	13.5 ± 0.9
DH82 macrophages	>1.6 × 10^5^
J774 macrophages	>1.6 × 10^5^

**Table 2 pharmaceutics-16-00041-t002:** Delamanid SLNP weight/weight percentage formula.

Component	% (*w*/*w*)
Delamanid	0.0015
PHC	0.0025
Stearic acid	0.0152
Ethanol	3.240
CTAB	0.0320
Tween 80	0.0800
PBS	96.6290

## Data Availability

Data are available from the corresponding authors.

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
