# Peer review of "Solid Lipid Nanoparticles Enhancing the Leishmanicidal Activity of Delamanid"

_pharmaceutics, 2023, doi:10.3390/pharmaceutics16010041_

Round 1

Reviewer 1 Report

Comments and Suggestions for Authors

The paper by Santamaria-Aguirre et al. describes the formulation of solid lipid nanoparticles (SLNPs) for the delivery of delamanid in the treatment of leishmaniasis. This includes the selection of delamanid among different drug candidates by computational methods, its incorporation into SLNPs, and the evaluation of its effectiveness against Leishmania parasites.

The results are interesting, and the information provided can be useful for numerous researchers, as leishmaniasis, a neglected tropical disease, remains a serious health problem, and new and more accessible drugs are urgently needed.

This study refers to an aspect that is very often neglected: the design of medicines that are both effective and accessible to the target population, in this case, the impoverished populations with limited access to healthcare that need to be treated for leishmaniasis. The solid lipid nanoparticles reported would become effective and accessible nanomedicines for the treatment of leishmaniasis.

Although the results are interesting, I would recommend a thorough revision of the manuscript before publication. Here are some points to be improved.

I would recommend rewriting Section 3.1 and describing the process leading to the selection of delamanid in more detail. In its present form, I find it difficult for the reader to follow, even when accessing the Supplementary Material.

The quality of Figures 1 and 2 should be improved to meet publication standards. Figure 1 should include a scale bar in the image to provide a visual reference for the size of the observed structures.

Table 1: Standard deviations should be included to provide insights into the precision, reliability, and statistical significance of the experimental IC50 data. The legend should provide additional information (number of determinations…).

References should be thoroughly reviewed. Verify that all entries follow the same format (e.g., abbreviated or full publication name) and that each reference entry includes all required elements as per the chosen style guide (check for missing volume or page numbers).

Reference 14, cited under Materials and Methods, section 2.1.2, and references 19, 20, and 21 cited under Results, section 3.2.1, correspond to Thesis reports written in Spanish. I strongly recommend incorporating the pertinent information from these references into the manuscript as Supplementary material. This will greatly facilitate easier access for most readers. Reference 34 is not cited in the text.

Author Response

Manuscript ID: pharmaceutics-2731988  

We are grateful for the time and effort that the reviewers have invested in assessing our manuscript. Their feedback has offered valuable insights that have contributed to refining the work. We hope our responses to their criticisms will be satisfactory and lead to the acceptance of the study for publication in Pharmaceutics.

REVIEWER #1:

The paper by Santamaria-Aguirre et al. describes the formulation of solid lipid nanoparticles (SLNPs) for the delivery of delamanid in the treatment of leishmaniasis. This includes the selection of delamanid among different drug candidates by computational methods, its incorporation into SLNPs, and the evaluation of its effectiveness against Leishmania parasites.

The results are interesting, and the information provided can be useful for numerous researchers, as leishmaniasis, a neglected tropical disease, remains a serious health problem, and new and more accessible drugs are urgently needed.

This study refers to an aspect that is very often neglected: the design of medicines that are both effective and accessible to the target population, in this case, the impoverished populations with limited access to healthcare that need to be treated for leishmaniasis. The solid lipid nanoparticles reported would become effective and accessible nanomedicines for the treatment of leishmaniasis. Although the results are interesting, I would recommend a thorough revision of the manuscript before publication. Here are some points to be improved.

  1. I would recommend rewriting Section 3.1 and describing the process leading to the selection of delamanid in more detail. In its present form, I find it difficult for the reader to follow, even when accessing the Supplementary Material.

Reply to referee.

We thank the reviewer. The text has now been rewritten.

  1. Table 1: Standard deviations should be included to provide insights into the precision, reliability, and statistical significance of the experimental IC50 data. The legend should provide additional information (number of determinations…).

Reply to referee.

The SD have now been included.

  1. References should be thoroughly reviewed. Verify that all entries follow the same format (e.g., abbreviated or full publication name) and that each reference entry includes all required elements as per the chosen style guide (check for missing volume or page numbers).

Reply to referee.

The References have been checked and revised.

  1. Reference 14, cited under Materials and Methods, section 2.1.2, and references 19, 20, and 21 cited under Results, section 3.2.1, correspond to Thesis reports written in Spanish. I strongly recommend incorporating the pertinent information from these references into the manuscript as Supplementary material. This will greatly facilitate easier access for most readers. Reference 34 is not cited in the text.

Reply to referee.

The Reference has been modified as indicated. Reference 34 is now cited.

Reviewer 2 Report

Comments and Suggestions for Authors

This article present the synthesis and characterization of solid lipid nanoparticles and their lieshmanicidal activity in vitro and in cellulo against amastigote forms.

The work is well structured and it shows exhaustive information about the leishmanicidal power of delamanid and derived SLPNs for the scientific community.

But there are several aspects to improve before consider it for publication:

- The number of references in the introduction section must be extended.

- I recommend include delamanid structure in the main text, not only in supplementary section.

- Delamanid sometimes appears in capital letter and others no. I reccomend to unify as "delamanid".

- Clarify the exact quantities of reagents employed for the synthesys of SLNPs for replication.

- SLNPs characterization section is too short. It must be extended adding more information about different pH stability, composition and textural properties.

I recommend accept after minor revision.

Author Response

Manuscript ID: pharmaceutics-2731988  

We are grateful for the time and effort that the reviewers have invested in assessing our manuscript. Their feedback has offered valuable insights that have contributed to refining the work. We hope our responses to their criticisms will be satisfactory and lead to the acceptance of the study for publication in Pharmaceutics.

REVIEWER #2:

The manuscript “Solid lipid nanoparticles enhancing the leishmanicidal activity of Delamanid” presented by Javier Santamaría-Aguirre and co-authors is a study aimed at developing a new formulation of Delamanid against leishmania. The authors applied bioinformatics and molecular docking methods to select the most effective anti-Leishmania drug from databases, and the results obtained confirmed the correctness of the choice, as well as the productivity of using this modern strategy.

The work is interesting; however, the manuscript needs improvement. In the Introduction, the authors convincingly show the need to create new drugs against leishmaniasis and justify the advisability of using nanoforms to improve the quality of previously rejected drugs. The goal of the work is justified and clearly stated.

Reply to referee.

We appreciate the reviewer for their constructive criticism and assistance in enhancing the manuscript.

  1. The number of references in the introduction section must be extended.

Reply to referee. I appreciate the opportunity to provide an update. I have extended the number of references in the manuscript, specifically incorporating an additional 19 references into the introduction. This revision aims to enrich the context and depth of the content.

  1. I recommend include delamanid structure in the main text, not only in supplementary section.(estaba incluido, en la figura 1).

Reply to referee. We thak the reviewer. It's worth noting that the structure of delamanid has been included in Figure 1 from the outset.

  1. Delamanid sometimes appears in capital letter and others no. I reccomend to unify as "delamanid.

Reply to referee. Thank you for the note. It's noted that the capitalization of delamanid has now been standardized. 

  1. Clarify the exact quantities of reagents employed for the syntheses of SLNPs for replication.

Reply to referee. Thank you for the update. I acknowledge that the percentages of the reagents in the formulation have been included in Table 2.

  1. SLNPs characterization section is too short. It must be extended adding more information about different pH stability, composition, and textural properties.

Reply to referee. Thank you for your comment. Regarding the dispersion of nanoparticles, it's important to note that the pH was consistently maintained at 7.4 throughout the experimentation. The composition details have already been included in the manuscript. As for the texture analysis, I would like to clarify that no specific texture assays were conducted in this study. However, we acknowledge the importance of atomic force microscopy (AFM) in providing valuable insights into the nanoscale surface characteristics. We agree with your suggestion and plan to incorporate AFM analysis in future investigations to enhance the comprehensiveness of our research.

Reviewer 3 Report

Comments and Suggestions for Authors

The document is well structured and the writing is appropriate, for my part this version can be published

Comments on the Quality of English Language

The document has minor errors in the language, they can be reviewed to improve

Author Response

Manuscript ID: pharmaceutics-2731988  

We are grateful for the time and effort that the reviewers have invested in assessing our manuscript. Their feedback has offered valuable insights that have contributed to refining the work. We hope our responses to their criticisms will be satisfactory and lead to the acceptance of the study for publication in Pharmaceutics.

REVIEWER #3:

We sincerely appreciate the reviewer for finding our paper suitable for publication and finding it acceptable in its current form. Thank you for your thorough review and thoughtful engagement with our work.

Reviewer 4 Report

Comments and Suggestions for Authors

 The manuscript “Solid lipid nanoparticles enhancing the leishmanicidal activity of Delamanid” presented by Javier Santamaría-Aguirre and co-authors is a study aimed at developing a new formulation of Delamanid against leishmania. The authors applied bioinformatics and molecular docking methods to select the most effective anti-Leishmania drug from databases, and the results obtained confirmed the correctness of the choice, as well as the productivity of using this modern strategy.

The work is interesting; however, the manuscript needs improvement.  

               In the Introduction, the authors convincingly show the need to create new drugs against leishmaniasis and justify the advisability of using nanoforms to improve the quality of previously rejected drugs. The goal of the work is justified and clearly stated.

Materials and Methods provides an overview of the methods used. The section is written somewhat carelessly, below are comments.

Line 145: please, insert time of sonication at 130 W / 20 KHz.

Line 148: Although magnetic particles are commercially available, at least their size characteristics must be given.

Line 154: Please, provide full characteristics of transmission electron microscope, including type of digital camera. Clarify the method of samples preparation for TEM study: did you contrast the NPs? etc.  

line 160: time of “gentle agitation”?

Please, provide information about amount of counted cells for evaluation the NPs size ( Section 2.3.2; Figures S4, S5).

Please, provide information about the approve of mice studies by appropriate Bioethics Committee.

Section 2.4.4. “Leishmanicidal activity in amastigotes” should contain the information about amounts of counted cells and amastigotes, and statistical evaluation of the differences.

It is necessary to present the characteristics of the macrophage cell lines used in the work and description of their cultivation.  

Please, check carefully all abbreviations and sense of the sentences – some words seem missed.

Results describe the work performed and the data obtained.

Comments:

Please, add to Figure 1 a photograph of intact promastigotes (control) similar to Figure 1b.

Please, add sample preparation characteristics for TEM in Legend of Figure 2 (d) TEM image of the SLNPs.

I cannot agree with the interpretation of the data in the section 3.4.2 “Subcellular Distribution of SLNPs in Infected Macrophages”.

First, the spherical structures in inserts 5a and 5b are designated as SLNPs. However, the structures have sizes of about 350 and 400 nm, respectively, which does not correspond to the previously determined size of no more than 200 nm, and in SLNPs presented on Fig. 2 have sizes – 100-110 nm. Such differences are too large to be explained as a result of different processing of the samples. Please, provide additional data evidencing that structures in inserts are SLNPs. 

The image of the dead parasite is also inconclusive - the structure is shown at very low magnification and its morphology and size does not resemble an amastigote.

The photographs shown in Fig. 4a do not demonstrate the penetration of NLNPs into macrophages, as well as the presence of amastigotes in macrophages. I think that authors used incorrect experimental design. A sufficient number of easily accessible TEM studies of Leishmania and its reproduction in cells have been published; the authors can be advised to refer to these works.

So, I suggest that authors remove the section on electron microscopy unless they have indisputable illustrations of SLNP penetration into macrophages and their effect on amastigotes present in the cells. Obviously, any TEM study requires careful preparation is very labor-intensive. Researchers conducting TEM must be experienced in identifying cellular structures and understand that a series of images of the structure must be obtained to accurately determine it. Examination of parasite-host interaction needs sampling at different time-points.

Figure 4b demonstrates data obtained by another method. I cannot comment this result, because method of Infection Index evaluation is not presented in Section 2.4.4. Leishmanicidal activity in amastigotes. No data on amount of the cells and amastigotes are in the manuscript. 

 Other sections of the paper did not cause the comments.

Comments on the Quality of English Language

In general, the text is written in understandable language. However, careful checking of the text, including abbreviations, is required. There are few incomprehensible phrases, apparently due to missing words.

Author Response

Manuscript ID: pharmaceutics-2731988  

We are grateful for the time and effort that the reviewers have invested in assessing our manuscript. Their feedback has offered valuable insights that have contributed to refining the work. We hope our responses to their criticisms will be satisfactory and lead to the acceptance of the study for publication in Pharmaceutics.

REVIEWER #4:

The manuscript “Solid lipid nanoparticles enhancing the leishmanicidal activity of Delamanid” presented by Javier Santamaría-Aguirre and co-authors is a study aimed at developing a new formulation of Delamanid against leishmania. The authors applied bioinformatics and molecular docking methods to select the most effective anti-Leishmania drug from databases, and the results obtained confirmed the correctness of the choice, as well as the productivity of using this modern strategy. The work is interesting; however, the manuscript needs improvement. In the Introduction, the authors convincingly show the need to create new drugs against leishmaniasis and justify the advisability of using nanoforms to improve the quality of previously rejected drugs. The goal of the work is justified and clearly stated.

Reply to referee. We sincerely appreciate the reviewer for their constructive comments and positive feedback. Your insightful observations on our manuscript have been invaluable, contributing significantly to the improvement of the text and enhancing the clarity of ideas within. Thank you for your thorough review and thoughtful engagement with our work.

  1. The goal of the work is justified and clearly stated.

Reply to referee. In addressing the aim of the work, it's important to note that in the summary of the study, specifically in sentence number 20, we explicitly state the vision of the study: "This study aims to repurpose approved drugs for an affordable leishmaniasis treatment."

  1. Materials and Methods provides an overview of the methods used. The section is written somewhat carelessly, below are comments.

Reply to referee. We appreciate the feedback from the reviewer, and in alignment with their suggestions, we have revised the Materials and Methods section. Additional information has been included to enhance the clarity and overall quality of the writing.

  1. Clarify the method of samples preparation for TEM study: did you contrast the NPs?

Reply to referee. We are grateful for the reviewer's feedback. The details regarding the preparation of TEM ultrathin sections and counterstaining are thoroughly described in section 2.4.2.

"2.4.2. Electron microscopy imaging of infected macrophages. Canine macrophage DH82, cultured in RPMI (RPMI Medium 1640 (1X) + Glutamax-l. Gibco. Ref. 61870-010) supplemented with 10% FBS and 0.05% Gentamicin, and incubated at 37 ºC / 5% CO2 were infected with L. infantum promastigotes for 4 hours. Then, non-internalized parasites were removed and cells were treated. After 12 hours, treatments (drug or SLNPs) at a concentration corresponding to the IC50 were added and incubated for 48 hours. After treatment, the cells were fixed with glutaraldehyde 3% for inclusion and post-fixed in 2% buffered osmium tetroxide, dehydrated in a graded acetone series, and embedded in Araldite resin (Durcupan ACM, Sigma-Aldrich). Ultrathin sections of ca. 70 nm thick, were obtained on an LKB ultramicrotome, stained with lead citrate and uranyl acetate. Imaging was performed using a JEOL JEM 1011 operated at 100 kV."

The SLNPs were not contrasted; they contain iron nanoparticles of approximately 10 nm diameter inside to enhance their TEM contrast, as demonstrated in our previous publications (https://doi.org/10.1016/j.bioactmat.2021.06.009), specifically in figures 1 and 2. Without this contrast agent, they are challenging to locate.

  1. Please, provide information about amount of counted cells for evaluation the NPs size ( Section 2.3.2; Figures S4, S5)

Reply to referee. Thank you for your comment. We would like to clarify that in our study, cell measurements were not conducted; rather, our focus was on nanoparticle measurements.

  1. Please, add sample preparation characteristics for TEM in Legend of Figure 2 (d) TEM image of the SLNPs.

Reply to referee. Thank you for your comment. A description of the preparation process has been added to the figure legend.

  1. I cannot agree with the interpretation of the data in the section 3.4.2 “Subcellular Distribution of SLNPs in Infected Macrophages”. First, the spherical structures in inserts 5a and 5b are designated as SLNPs. However, the structures have sizes of about 350 and 400 nm, respectively, which does not correspond to the previously determined size of no more than 200 nm, and in SLNPs presented on Fig. 2 have sizes – 100-110 nm. Such differences are too large to be explained as a result of different processing of the samples. Please, provide additional data evidencing that structures in inserts are SLNPs. The image of the dead parasite is also inconclusive - the structure is shown at very low magnification and its morphology and size does not resemble an amastigote. The photographs shown in Fig. 4a do not demonstrate the penetration of SLNPs into macrophages, as well as the presence of amastigotes in macrophages. I think that authors used an incorrect experimental design. A sufficient number of easily accessible TEM studies of Leishmania and its reproduction in cells have been published; the authors can be advised to refer to these works. So, I suggest that authors remove the section on electron microscopy unless they have indisputable illustrations of SLNP penetration into macrophages and their effect on amastigotes present in the cells. Obviously, any TEM study requires careful preparation is very labor-intensive. Researchers conducting TEM must be experienced in identifying cellular structures and understand that a series of images of the structure must be obtained to accurately determine it. Examination of parasite-host interaction needs sampling at different time-points.

Reply to referee. We would like to emphasize our considerable expertise in the field of transmission electron microscopy (TEM) studies of nanoparticles at the intracellular level. Throughout our research journey, we have contributed to numerous publications utilizing TEM to elucidate the subcellular localization of various nanomaterials, ranging from carbon nanotubes to gold/iron nanoparticles and solid-lipid nanoparticles. For your convenience, we have compiled a bibliography webpage showcasing our recent publications, which can be explored here: https://mlfanarraga.wixsite.com/grupo-nanomedicina/recent-publications.

We appreciate and acknowledge your valid point concerning potential distortions in nanoparticle size, which we believe could be attributed to the time spent intracellularly and the associated chemical changes in the lipid structure. Additionally, we recognize, as the referee is likely aware, that image processing for electron microscopy, involving sample dehydration, can significantly impact the lipid composition of nanoparticles, as we have observed in our previous works.

However, although we acknowledge the inherent limitations of any methodology, we firmly believe that the TEM ultramicrographs presented in our manuscript are inherently elucidating. Collectively, they provide compelling evidence supporting the presence of nanoparticles inside macrophages. This conviction is further supported by the identification of iron nanoparticles (of 10 nm average) which had been included in the formulation of the SLNPs for their TEM identification. We have successfully employed this identification method in the analysis of other wax solid-lipid nanoparticles produced and investigated in our laboratory. (https://doi.org/10.1016/j.bioactmat.2021.06.009).

Regarding the identification of the structures inside cells, we agree with your caution. While we cannot provide a guarantee that these structures are parasites, as we have not performed immunostaining or any other direct identification technique, these macrophages were only incubated with parasites. The observed features, such as a compacted and electron-dense nucleus imaging indicating a high probability of deoxyribonucleic acid (DNA), strongly suggest the structure corresponds to a parasite since no other macrophage structure looks like this. Given our extensive experience and the consistent patterns observed in numerous images of infected cells, the likelihood that these are cells infected by leishmania parasites is significant.

Hence, we genuinely appreciate your thoughtful comments but we have decided to retain the TEM figure. To comply with the claims, we have now modified it by incorporating additional images of an infected cell, showcasing another cytoplasmic SLNP. Like the previous case, the iron nanoparticles within the SLNPs play a crucial role in unequivocally identifying the SLNP. In this second image, the parasite is much more clearly depicted, displaying a patent flagellum that can be distinctly observed inside the phagosome membrane.

Also, in response to your concerns, we are open to adjusting the text in the figure caption, adopting a less categorical stance by acknowledging the possibility that the observed structures could be nanoparticles. Your valuable feedback is appreciated, and if there are specific areas you believe require further clarification or if additional data would be beneficial, please do not hesitate to inform us. Your insightful review has played a key role in refining the quality of our work.

  1. All these minor comments have been appropriately changed/amended in the manuscript text. Thank you very much for these valuable observations.
  • Line 145: please, insert time of sonication at 130 W / 20 KHz.
  • Line 148: Although magnetic particles are commercially available, at least their size characteristics must be given.
  • Line 154: Please, provide full characteristics of transmission electron microscope, including type of digital camera.
  • line 160: time of “gentle agitation”
  • Please, provide information about the approval of mice studies by appropriate Bioethics Committee.
  • Section 2.4.4. “Leishmanicidal activity in amastigotes” should contain the information about amounts of counted cells and amastigotes, and statistical evaluation of the differences.
  • It is necessary to present the characteristics of the macrophage cell lines used in the work and description of their cultivation.
  • Please, check carefully all abbreviations and sense of the sentences – some words seem missed.
  • Please, add to Figure 1 a photograph of intact promastigotes (control) similar to Figure 1b.
  • Figure 4b demonstrates data obtained by another method. I cannot comment this result, because method of Infection Index evaluation is not presented in Section 2.4.4. Leishmanicidal activity in amastigotes.
  • No data on the amount of the cells and amastigotes are in the manuscript.

Round 2

Reviewer 1 Report

Comments and Suggestions for Authors

The manuscript has been significantly improved; however, some points still need to be modified before publication. I strongly recommend that authors follow the instructions given in the first review of the manuscript:

References should be thoroughly reviewed to verify that all entries follow the same format (e.g., abbreviated or full publication name) and that each reference entry includes all required elements as per the chosen style guide (check for missing volume or page numbers). Many of the references still need to be modified.

Some of the references (e.g., 33, 34, 35, 39, 40, and 41) still correspond to reports written in Spanish. I strongly recommend that, if possible, authors replace them with alternative references in English to facilitate easier access for most readers.

Author Response

We sincerely appreciate the reviewer for taking the time to conduct a second review of our manuscript. We want to emphasize that we have indeed incorporated all the modifications requested during the initial review. However, it is important to note that, in accordance with the reference format specified by this journal, references must be included exactly as they have been written. Nevertheless, should there be any defect in the reference format, it will undoubtedly be rectified by the publisher.

We would like to inform the reviewer that references in English equivalent to those discussed in Spanish have been included. However, as this is an original work, we believe it is appropriate to cite them in their original published form.

Reviewer 4 Report

Comments and Suggestions for Authors

Review attached, see file

Comments on the Quality of English Language

Some polishing of English would be useful.

Author Response

We appreciate the reviewer's feedback on the revised version of our work. It's important to note that, following previous suggestions, we have subjected the entire text to grammatical corrections using two different software tools. This process ensures grammatical accuracy and a lack of errors. Indeed, the quality and finesse of the final English can be subjective and vary depending on the reader. Nevertheless, we sincerely appreciate the reviewer for their valuable suggestion.